Profiling exercise intensity during the exergame Hollywood Workout on XBOX 360 Kinect®

Viana Ricardo B. 1
Vancini Rodrigo L. 2
http://orcid.org/0000-0002-0083-2910 Vieira Carlos A. 1
http://orcid.org/0000-0003-2459-4977 Gentil Paulo 1
http://orcid.org/0000-0002-4724-2221 Campos Mário H. 1
Andrade Marilia S. 3
http://orcid.org/0000-0001-5749-6877 de Lira Claudio Andre B. 1 andre.claudio@gmail.com
1 Faculdade de Educação Física e Dança, Universidade Federal de Goiás , Goiânia , Goiás , Brazil
2 Centro de Educação Física e Desportos, Universidade Federal do Espírito Santo , Vitória , Espírito Santo , Brazil
3 Departamento de Fisiologia, Universidade Federal de São Paulo , São Paulo , São Paulo , Brazil
Ramírez-Campillo Rodrigo
Electronic publication date: 2018 Aug 30
Publication date: 2018
Volume: 6
Electronic Location ID: e5574
Received 2018 Jun 3; Accepted 2018 Aug 14
Copyright: © 2018 Viana et al.
Copyright year: 2018
Copyright holder: Viana et al.
License: This is an open access article distributed under the terms of the Creative Commons Attribution License, which permits unrestricted use, distribution, reproduction and adaptation in any medium and for any purpose provided that it is properly attributed. For attribution, the original author(s), title, publication source (PeerJ) and either DOI or URL of the article must be cited.
License URL: https://creativecommons.org/licenses/by/4.0/

Keywords: Physiological responses, Oxygen uptake, Heart rate

Funding: Fundação de Amparo à Pesquisa do Estado de Goiás-FAPEG/Brazil 201210267001056 Conselho Nacional de Desenvolvimento Científico e Tecnológico-CNPq/Brazil 405096/2016-0 This study was funded by Fundação de Amparo à Pesquisa do Estado de Goiás-FAPEG/Brazil (grant number 201210267001056) and by Conselho Nacional de Desenvolvimento Científico e Tecnológico-CNPq/Brazil (grant number 405096/2016-0). The funders had no role in study design, data collection and analysis, decision to publish, or preparation of the manuscript.

==============================
Background

Despite the increasing popularity of exergame practice and its promising benefits in counteracting physical inactivity, limited research has been performed to document the physiological responses during an exergame session. This study aims (i) to investigate the responses of heart rate (HR) and oxygen uptake (V˙O2) during an exergame session and to compare with HR and V˙O2 measured during joystick session and (ii) to compare HR and V˙O2 obtained during exergame and joystick session with those HR and V˙O2 associated with first and second ventilatory thresholds (VT1 and VT2, respectively) obtained during a maximal graded exercise test.

Methods

A total of 39 participants performed a maximal graded exercise test to determine maximal oxygen uptake (V˙O2max), VT1, and VT2. On separate days, participants performed an exergame and traditional sedentary game (with a joystick) sessions. The time that participants remained with HR and V˙O2 below the VT1, between the VT1 and VT2 and above the VT2 were calculated to determine exercise intensity.

Results

Heart rate and V˙O2 were below VT1 during 1,503 ± 292 s (86.1 ± 16.7%) and 1,610 ± 215 s (92.2 ± 12.3%), respectively. There was an increase in HR and V˙O2 as a function of exergame phases, since HR mean values in the ‘warm-up’ period (119 ± 13 bpm) were lower than the ‘main phase’ (136 ± 15 bpm) and ‘cool-down’ periods (143 ± 15 bpm) (p < 0.001). Regarding V˙O2 values, the ‘warm-up’ (25.7 ± 2.9 mL.kg−1.min−1) were similar to the ‘main phase’ (25.1 ± 2.8 mL.kg−1.min−1) (p > 0.05) and lower than the ‘cool-down’ (28.0 ± 4.8 mL.kg−1.min−1) (p < 0.001). For all times of the joystick session, average HR and V˙O2 were below the VT1 levels.

Conclusion

Exergames can be classified as light to moderate exercise. Thus, exergames could be an interesting alternative to traditional forms of exercise.

Introduction

Physical inactivity is associated with an increase in the risk of a variety of chronic diseases (such as diabetes mellitus and arterial hypertension) and consequently, premature deaths (Ding et al., 2016). Although well documented, a large part of the population remains sedentary (Fox, 2012). In this context, adults who are not engaged in traditional exercise methods with a minimum intensity corresponding to 55% of maximal heart rate (HRmax) require a strategy to achieve and maintain sufficient physical activity for health benefits (Garber et al., 2011).

Exergames, or active videogames, may motivate some adults to engage in physical activity and may be an attractive and fun alternative of physical activity for increasing the motivation of people to engage in an exercise programme (Street, Lacey & Langdon, 2017).

Exergames can be defined as electronic games that allow players to physically interact with images on screen (Foley & Maddison, 2010). Traditionally, exergames are characterized as physical activities including games that simulate walking, running, stair climbing, cycling, rowing, and swimming (Graves et al., 2010; Wu, Wu & Chu, 2015). Furthermore, there are games that require mixed activity that simulate jumping, throwing, and kicking (Bird et al., 2015).

Most studies involving exergames investigated the acute and chronic effects on caloric expenditure (Barkman et al., 2016), motor aspects (Collado-Mateo et al., 2017), and cognitive function (Garcia et al., 2016; Stanmore et al., 2017) in different healthy and clinical populations (Dos Santos Mendes et al., 2012; Collado-Mateo et al., 2017). However, less is known about the physiological responses during an exergame session, especially regarding oxygen uptake (V˙O2) and heart rate (HR). These physiological variables are a common form for gathering information about exercise intensity. Moreover, these physiological variables are useful tools to monitor responses and adaptations to exercise. In this regard, it has previously been shown that an increase in one metabolic equivalent in exercise capacity decreases mortality rate by 12% (Myers et al., 2002; Kodama et al., 2009).

Neves et al. (2015) investigated the acute cardiovascular responses during a session of the game Zumba Fitness Core® performed with XBOX 360 and observed a significant increase of HR immediately after the session. Graves et al. (2010) compared the physiological cost among adolescents, young, and older adults during a game with a joystick, Wii® Fit activities (yoga, muscle conditioning, balance, and aerobics), and brisk treadmill walking and jogging and found that energy expenditure and HR of Wii® Fit activities were greater than joystick games, but lower than treadmill exercise and that the Wii® elicited moderate intensity activity. Rodrigues et al. (2015) evaluated the acute metabolic and cardiovascular responses of healthy men during exergames with the Nintendo Wii® (obstacle course, hula hoop, free run, soccer heading, penguin slide, and table tilt) and found that the Wii® exercises are considered to be of light and moderate intensity. Wu, Wu & Chu (2015) examined and compared the energy expenditure and intensity of XBOX 360 Kinect® exergames (boxing, soccer, track and field, ping pong, beach volleyball, and bowling) in healthy young adults and observed that boxing and soccer exergames provided the greater exercise intensity. It is important to note that none of these studies determined exercise intensity based on metabolic thresholds obtained in a continuous laboratory treadmill test, such as maximal graded exercise testing (GXT) (De Lira et al., 2013).

Thus, despite the increasing popularity of exergame practice (Sween et al., 2014) and its promising benefits in counteracting physical inactivity, limited research has been performed to document the physiological responses during an exergame session, which makes it difficult to characterize the physiological responses and brings uncertainty regarding its potential benefits in increasing physical fitness. Therefore, the aims of this study were to investigate physiological parameters of young men, to describe HR and V˙O2 responses during the exergame Hollywood Workout on XBOX 360 Kinect session, and to compare with responses in a joystick game. Also, the study compared HR and V˙O2 responses during exergame with those HR and V˙O2 associated with first and second ventilatory thresholds (VT1 and VT2, respectively) attained during a maximal graded exercise test.

The exergame Hollywood Workout on XBOX 360 Kinect was chosen with the intention of maintaining ‘ecological validity,’ as this game simulates some exercises, such as, push-ups, skater lunges, and bicycle crunches that are performed in exercise facilities.

Materials and Methods

Participants

A total of 39 young men were recruited among students from the Faculty of Physical Education and Dance of the Federal University of Goiás (Brazil). Participants (25.9 ± 4.9 years, 1.79 ± 0.08 m, 79.2 ± 13.8 kg, 24.7 ± 3.4 kg.m−2) were recruited through social media and direct contact. Participants were physically active, asymptomatic and non-users of exergames. All participants were informed of the intent, experimental procedures, benefits, and risks of the study and informed consent was obtained from all individual participants included in the study. All experimental procedures were approved by the University Human Research Ethics Committee (no 1.459.010) and conformed to the principles outlined in the Declaration of Helsinki.

Baseline examinations

Before the beginning of the experiment, each participant came to the laboratory for anamnesis through the Physical Activity Readiness Questionnaire (PAR-Q). The inclusion criteria were to respond ‘no’ on all the PAR-Q questions. No participants were excluded.

Study design

Each participant reported to the laboratory on three separate days. The first day involved baseline examinations and GXT, and the other 2 days consisted of two randomly ordered sessions (exergame and joystick). The HR and V˙O2 corresponding to VT1 and VT2 obtained in GXT were used as analysis parameters for the HR and V˙O2 recorded during the exergame and joystick sessions. All sessions were started at the same time of day with at least 48 h separating the beginning of each visit. Participants were instructed to eat a standardized meal, not to participate in any strenuous exercise, and not to consume any stimulant or alcohol in the 24 h preceding all testing sessions. The temperature and relative humidity in the testing laboratory ranged from 21 to 23 °C and 55% to 65%, respectively, for all trials.

Experimental procedures

Maximal graded exercise testing

Graded exercise testing was administrated to determine VT1, VT2, and maximal oxygen uptake (V˙O2max), as well as their associated running velocities, HR and V˙O2. Prior to performing the GXT, participants were given a standardized set of instructions explaining the test. On completion of these preliminary procedures, each participant underwent an incremental maximal exercise test on a motorized treadmill (ATL, Inbramed, Porto Alegre, Brazil) with 0% slope. The schedule of this test consisted of a 5-min warm-up period at seven km.h−1, and then the initial speed was progressively increased by one km.h−1 every minute until exhaustion (De Lira et al., 2013). During the exercise testing, participants were verbally encouraged to exercise for as long as possible. Respiratory gas samples were measured continuously using a metabolic system (VO2000; MedGraphics, Saint Paul, USA). Prior to testing, the metabolic system was calibrated according to the manufacturer’s instructions. HR was recorded using a HR-monitor (RS800CX; Polar Electro, Espoo, Kempele, Finland). Figure 1 shows a participant during the GXT.

Figure 1 Maximal graded exercise testing session.

Photo credit: Ricardo B. Viana.

The following data (averaged over 10 s) were obtained: V˙O2 (mL.kg−1.min−1) at standard temperature (0 °C) and barometric pressure at sea level, carbon dioxide production (V˙CO2) (mL.kg−1.min−1) at standard temperature (0 °C) and barometric pressure at sea level, respiratory exchange ratio (RER), minute ventilation (V˙E) (L.min−1) at body temperature and saturation pressure, respiratory rate (breaths per minute (bpm)), ventilatory equivalents for O2 and CO2 (V˙E/V˙O2 and V˙E/V˙CO2, respectively), expired fractions of O2 and CO2 (%) and HR (beats per minute (bpm)). Peak treadmill speed was defined as the last achieved running speed sustained for at least 30 s. V˙O2max was defined as the highest 10-s averaged V˙O2 value with inclusion criteria consistent with conventional guidelines for V˙O2max (e.g. an inability to sustain the workload, relative HR > 95% predicted for their age, RER at maximal exercise ≥ 1.1, and V˙O2 plateau [the point at which V˙O2 increases less than 150 mL.min−1 for a given increase in workload]) (Howley, 2007). VT1 and VT2 were assessed using established criteria (Wasserman et al., 2005). Briefly, VT1 corresponds to the break point in the plot of V˙CO2 as a function of V˙O2. At that point, V˙E/V˙O2 increases without an increase in V˙E/V˙CO2. VT2 was located between VT1 and V˙O2max, when V˙E/V˙CO2 starts to increase and V˙E/V˙O2 continues to increase. VT1 and VT2 were determined independently by two experienced investigators. If agreement between the investigators was not achieved, VT1 and VT2 were determined by consensus. To determine the V˙O2 and V˙E at VT1 and VT2, the average of the last 10 s of each corresponding level was used. In practical terms, ventilatory thresholds represent points that can be used to classify the intensity of aerobic exercise.

Exergame session

The exergame session was conducted in a room (10 × 6 m) and was accompanied by a certified trainer that was experienced in exergames. For the purposes of the present study, the exergame Hollywood Workout (Majesco Entertainment, Edison, Hazlet, NJ, USA) was used with the intention of maintaining ‘ecological validity,’ as this game simulates some exercises usually performed in exercise facilities. Indeed, this exergame has several pre-established training protocols consisting of standardized exercises. Due to the feasibility of performing the exercises by coupling the participant to the metabolic system, the training protocol used in the present study was the Sports Athlete.

The Sports Athlete protocol consists of three phases: warm-up, main, and cool-down, with a total exercise time of 19 min and 28 s and total time in transitions between exercises of 8 min and 32 s, totalling a session of 28 min. However, total session time could vary according to the ability of the participant in the transitions between exercises. At the end of the Sports Athlete protocol, a numerical total score was provided. Table 1 provides the exercises used in Sports Athlete protocol.

Table 1 Sports athlete protocol of the exergame Hollywood Workout.

Phases	Exercises	Sets	Time (seconds)	Repetitions*	
Warm-up	Jog	1	71	25	
Trunk crosses	1	62	20	
Jumping jacks	1	68	40	
Ice skaters	1	59	12	
Main	Push-ups	1	59	12	
Skater lunges	1	69	16	
Bicycle crunches	1	74	25	
Push-ups	1	73	15	
Skater lunges	1	82	24	
Bicycle crunches	1	74	25	
Push-ups	1	74	15	
Skater lunges	1	81	24	
Bicycle crunches	1	74	25	
Cool-down	Punches	1	61	25	
Jump rope	1	56	40	
Side shuffles	1	68	20	
Mountain climbers	1	63	25	
Note:

* Repetitions proposed by the exergame. However, the participant was instructed to perform as many repetitions as possible.

Heart rate and V˙O2 were recorded before and during (including rest periods between sets) the Sports Athlete protocol. HR (5 s average HR value) and V˙O2 (10 s average V˙O2 value) registers were monitored by a HR-monitor (RS800CX; Polar Electronics, Finland) and metabolic system (VO2000; MedGraphics, USA), respectively. HR and V˙O2 recorded during the Sports Athlete protocol were compared with HR and V˙O2 corresponding to VT1 and VT2 obtained in GXT Fig. 2 shows participants performing the Sports Athlete protocol.

Figure 2 Some exercises of the Sports Athlete protocol performed by the participants during the exergame session.

(A) Skater lunges; (B) jog; (C) jump rope; (D) push-ups. Photo credit: Ricardo B. Viana.

Joystick session

The participants also underwent a 25-min joystick game session (traditional sedentary video gaming) (Fig. 3). To reduce the effect of playing ability among participants was chosen the joystick-game Injustice: Gods among us, ultimate edition. This game is easy to play and allow the participants restart the fight easily and faster. We tried other joystick-games (fight games) but the participants dispended a lot of time between one fight and other. Indeed, we did not find a joystick-game similar with the exergame Hollywood Workout, since it is an exergame which simulates traditional physical exercises. HR and V˙O2 values were recorded during all time sessions and compared with the HR and V˙O2 corresponding to VT1 and VT2 obtained in GXT.

Figure 3 Joystick session.

Photo credit: Ricardo B. Viana.

Statistical analysis

Age, the highest HR and percent of maximal heart rate (HRmax), and average V˙O2 in the exergame session presented a normal distribution (p > 0.05) according to the Shapiro–Wilk tests. All other variables in the exergame and joystick sessions presented a non-normal distribution (p < 0.05). Differences between the exergame and joystick sessions were analysed by the paired Student t-test (data with normal distribution) and Wilcoxon test (data with non-normal distribution). A one way repeated-measures analysis of variance was used to compare the differences between variables with normal distribution assessed in the ‘warm-up,’ ‘main phase,’ and ‘cool-down’ periods of the exergame session. When necessary, post hoc testing was performed by multiple comparisons using the Bonferroni procedure for confidence interval correction. The Friedman test was used to compare the differences between variables with non-normal distribution assessed in the ‘warm-up,’ ‘main phase,’ and ‘cool-down’ periods of the exergame session. The measures of the effect size for bilateral differences were calculated by dividing the mean difference by the standard deviation of the pre-training measurement. The magnitude of the effect sizes was judged according to the following criteria: d = 0.2 considered a ‘small’ effect size; 0.5 represented a ‘medium’ effect size; and 0.8 a ‘large’ effect size (Cohen, 1988). All statistical analyses were performed with the Statistical Package for the Social Sciences version 20.0. The significance level was set p < 0.05. Data are shown as the means ± standard deviations.

Results

Maximal graded exercise testing

Data from the GXT are shown in Table 2. VT1 and VT2 were detected in all cases.

Table 2 Physiological variables obtained by maximal graded exercise testing.

GXT (n = 39)	V˙O2 (mL.kg−1.min−1)	V.O2max	HR (bpm)	% Attained HRmax	Speed (km.h−1)	
VT1	38.1 ± 5.9	60.9 ± 6.3	156 ± 12	82.0 ± 4.9	10.6 ± 1.2	
VT2	48.7 ± 6.3	77.7 ± 5.7	172 ± 10	90.2 ± 3.7	13.2 ± 1.4	
Maximal exercise	62.7 ± 7.4	N/A	191 ± 11	N/A	16.1 ± 1.7	
Notes:

Data are presented as the mean ± standard deviation.

GXT, maximal graded exercise test; V˙O2, oxygen uptake; V˙O2max, maximal oxygen uptake; HR, heart rate; HRmax, maximal heart rate; VT1, first ventilatory threshold; VT2, second ventilatory threshold; bpm, beats per minute; N/A, not applicable.

Exergame session

Heart rate behaviour during the Sports Athlete protocol is shown in Table 3. The highest HR and percentage of HRmax attained in both the ‘main phase’ and ‘cool-down’ of the exergame session were about 11 ± 10%, higher than those attained in the ‘warm-up’ (p < 0.001 for both, effect size = 1.17–1.30 [large] for all comparisons). The average HR and percentage of HRmax attained in the ‘cool-down’ was 20.4 ± 11.8% and 5.2 ± 7.1%, higher than those attained in the ‘warm-up’ (p < 0.001, effect size = 1.71 and 1.76 [large], respectively) and ‘main phase’ (p < 0.001, effect size = 0.47 [small] and 0.51 [medium], respectively), respectively. The average HR and percentage of HRmax attained in the ‘main phase’ was about 14.6 ± 9.7%, higher than that attained in the ‘warm-up’ (p < 0.001, effect size = 1.21 and 1.33 [large], respectively). During the ‘warm-up,’ the average time spent at a HR below the VT1, between the VT1 and VT2, and above the VT2 was 289 ± 27 s, 10 ± 25 s, and 1 ± 4 s, respectively. In the ‘main phase,’ the average time spent at a HR below the VT1, between the VT1 and VT2, and above the VT2 was 947 ± 197 s, 119 ± 144 s, and 34 ± 87 s, respectively. During the ‘cool-down,’ the average time spent at a HR below the VT1, between the VT1 and VT2, and above the VT2 was 271 ± 83 s, 63 ± 66 s, and 14 ± 41 s.

Table 3 Heart rate, oxygen uptake and time (expressed as relative percentages) in each metabolic zone.

	Warm-up	Main phase	Cool-down	p	
HR	
Highest HR (bpm)	149 ± 12	165 ± 15*	164 ± 13*	<0.001	
% Attained HRmax	78.5 ± 6.3	86.5 ± 6.4*	86.4 ± 7.2*	<0.001	
Average HR (bpm)	119 ± 13	136 ± 15#	143 ± 15*#	<0.001	
% Attained HRmax	62.6 ± 6.7	71.4 ± 6.5#	74.9 ± 7.3*#	<0.001	
Time below to HR at VT1 (%)	96.5 ± 9.0	86.1 ± 17.9*	78.4 ± 24.1*	<0.001	
Time between to HR at VT1–VT2 (%)	3.3 ± 8.0	10.8 ± 13.1*	18.3 ± 19.2*	<0.001	
Time above to HR at VT2 (%)	0.2 ± 1.3	3.1 ± 7.9*	3.2 ± 9.8	0.015	
V˙O2	
Highest V˙O2 (mL·kg−1·min−1)	38.5 ± 4.5	38.3 ± 4.9	41.7 ± 7.5#	0.009	
% Attained V˙O2max	62.0 ± 8.9	61.7 ± 9.0	66.9 ± 12.7#	0.009	
Average V˙O2 (mL·kg−1·min−1)	25.7 ± 2.9	25.1 ± 2.8	28.0 ± 4.8*#	<0.001	
% Attained V˙O2max	41.5 ± 7.2	40.5 ± 6.9	45.3 ± 9.4#	<0.001	
Time below to V˙O2 at VT1 (%)	90.2 ± 14.7	95.3 ± 11.3*	84.0 ± 20.7#	0.001	
Time between to V˙O2 at VT1–VT2 (%)	9.0 ± 13.0	4.1 ± 10.0	12.0 ± 14.1#	0.024	
Time above to V˙O2 at VT2 (%)	0.8 ± 3.8	0.6 ± 3.4	4.0 ± 11.8	0.010	
Notes:

Heart rate, oxygen uptake and time (expressed as relative percentages) in each metabolic zone, defined as below VT1, between the VT1 and VT2, and above the VT2 during each phase of the exergame session (n = 39).

Data are presented as the means ± standard deviation.

bpm, beats per minute; HR, heart rate; HRmax, maximum heart rate; N/A, not applicable; V˙O2, oxygen uptake; VT1, first ventilatory threshold; VT2, second ventilatory threshold.

* Significant difference from ‘warm-up.’

# Significant difference from ‘main phase.’

The V˙O2 behaviour during the Sports Athlete protocol is also shown in Table 3. The highest V˙O2 attained in the ‘cool-down’ of the Sports Athlete protocol of the exergame Hollywood Workout was 8.8 ± 15.2%, higher than those attained in the ‘main phase’ (p = 0.009, effect size = 0.54 [medium]). The highest percent of V˙O2max attained in the ‘cool-down’ was 8.8 ± 15.2%, higher than those attained in the ‘main phase’ (p = 0.009, effect size = 0.47 [small]). The average V˙O2 attained in the ‘cool-down’ was 9.7 ± 16.9% and 11.5 ± 11.1%, higher than those attained in the ‘warm-up’ (p = 0.005, effect size = 0.58 [medium]) and ‘main phase’ (p < 0.001, effect size = 0.74 [medium]), respectively. The average percent of V˙O2max attained in the ‘cool-down’ was 11.5 ± 11.1%, higher than those attained in ‘main phase’ (p < 0.001, effect size = 0.36 [small]), respectively.

As compared with V˙O2 corresponding to VT1 and VT2, in the ‘warm-up,’ the average time spent at a V˙O2 below the VT1, between the VT1 and VT2, and above the VT2 was 271 ± 44 s, 27 ± 39 s, and 2 ± 11 s, respectively. In the ‘main phase,’ the average time spent below the VT1 V˙O2 between the VT1 and VT2, and above the VT2 was 1,048 ± 125 s, 45 ± 109 s, and 6 ± 37 s, respectively. Finally, in the ‘cool-down,’ the mean time spent below the VT1 V˙O2, between the VT1 and VT2, and above the VT2 was 294 ± 87 s, 42 ± 49 s, and 14 ± 43 s, respectively.

Overall, HR and V˙O2 of participants in the exergame session remained on average 1,503 ± 292 s (86.1 ± 16.7% of the total time session) and 1,610 ± 215 s (92.2 ± 12.3% of the total time session), respectively, below their VT1 values (Table 3).

The highest HR and percentage of HRmax in the exergame session were higher (96.4 ± 29.7% for both) than in the joystick session (p < 0.001 for both, effect size = 6.21 and 6.19 [large], respectively). Additionally, the average HR and percentage of HRmax in the exergame session were higher (83.1 ± 39.5 for both) than in the joystick session (p < 0.0001, effect size = 5.18 and 6.37 [large], respectively). Regarding V˙O2, the highest V˙O2 and percentage of V˙O2max in the exergame session were four times (401.4 ± 120.2% for both) higher than in the joystick session (p < 0.001, effect size = 6.64 and 6.34 [large], respectively). The average V˙O2 and percentage of V˙O2max in the exergame session were four times (401.7 ± 84.3% for both) higher than in the joystick session (p < 0.001, effect size = 9.26 and 6.48 [large], respectively). For all times of the joystick session (25 min), average HR (73 ± 9 bpm) and V˙O2 (5.3 ± 1.4 mL.kg−1.min−1) were spent below the VT1 levels, corresponding to 35.2 ± 5.0% of HRmax and 8.5 ± 1.8% of V˙O2max of the participants, respectively. The highest HR (88 ± 12 bpm) and V˙O2 values (9.2 ± 3.2 mL.kg−1.min−1) attained in the joystick session corresponded to 46.2 ± 6.9% of HRmax and 14.7 ± 4.6% of V˙O2max of the participants, respectively. All variables in the joystick session presented lower values to the exergame session (p < 0.001).

Discussion

The main aims of the present study were to investigate the responses of HR and V˙O2 in young men during an exergame session (Hollywood Workout on XBOX 360 Kinect®) and compare with HR and V˙O2 measured during joystick session. We also aimed to compare HR and V˙O2 during the exergame session in relation to VT1 and VT2. As expected, physiological response values during the joystick session were lower than those evaluated during the exergame session. In addition, we found that the average intensity of the Sports Athlete protocol was ‘light to moderate,’ as an appreciable percentage (∼86% [HR] and ∼92% [V˙O2]) of the exercise bout occurred at an intensity lower than HR and V˙O2 in VT1 levels (Stangier et al., 2016). Confirming this finding, the average percentage HRmax and percentage V˙O2max during the Sports Athlete protocol were 70.7% and 41.6%, respectively.

These values are similar to data from other studies with exergames (Tan et al., 2002; Unnithan, Houser & Fernhall, 2006; Jordan, Donne & Fletcher, 2011) and contrary to other (Neves et al., 2015). Jordan, Donne & Fletcher (2011) reported that 15 healthy men (age 29 ± 4 years, body mass 81 ± 12 kg, height 1.77 ± 0.05 m, body mass index 25.9 ± 3.8 kg.m−2, V˙O2peak 44.8 ± 5.5 ml.kg−1.min−1) attained ∼66% and ∼72% of HRmax and ∼41% and ∼56% of V˙O2max during Wii® boxing and an exergame controlled only with movements of the lower limbs through the PlayStation® 2, respectively. Tan et al. (2002) reported that 40 young individuals (21 men and 19 women, age 17.5 ± 0.7 years) attained an average HR of 137 bpm (70% of HRmax), average V˙O2 of 24.6 mL.kg−1.min−1 (44% of V˙O2max) during the exergame Dance Dance Revolution 3rd Mix™ Konami®. Unnithan, Houser & Fernhall (2006) reported that children and adolescents (11–17 years) attained ∼65% of HRmax during the exergame Dance Dance Revolution™ Konami®. On the other hand, average V˙O2 (∼35% of V˙O2peak) did not reach the minimum values for development and maintenance of cardiorespiratory fitness (>46% of V˙O2max) proposed by the American College of Sports Medicine (ACSM) (Garber et al., 2011), similar to the results of the present study. Mills et al. (2013) and Lau et al. (2015) found similar results to the present study regarding percentage of HRmax in children during a high-intensity exergame (Kinect Sports–200 m Hurdles) and the exergame I-Dong running, respectively; however, the percentage of V˙O2max reported by Mills et al. (2013) and Lau et al. (2015) was, respectively, lower (40.3%) and higher (61.4%) than that found in the present study. In general, despite previous studies investigating different populations and exercises, our results are in line with the literature. However, it is important to emphasize that the presented data should be extrapolated with caution.

According to the ACSM criteria, the intensity of the Sports Athlete protocol of the exergame Hollywood Workout can be classified as moderate, since it elicited average HR values corresponding to 70.7% of HRmax, framed in the range of 64–76% of HRmax proposed by the ACSM (Garber et al., 2011). The moderate intensity of the Sports Athlete protocol can be justified by fact that the evaluated participants presented high cardiorespiratory fitness, demanding a lower HR and V˙O2 for the same workload when compared to sedentary individuals (DeMaria et al., 1978). In addition, the ACSM suggests that exercise must have a minimum intensity corresponding to 55% of HRmax to evoke the benefits of aerobic training (Garber et al., 2011). Therefore, the Sports Athlete protocol of the exergame Hollywood Workout can be a useful tool for improving cardiorespiratory fitness as well as being an alternative tool to traditional exercise protocols. However, it should be noted that for this purpose, the weekly exercise volume of at least 150 min (which would correspond to five exercise sessions, considering that each Sports Athlete protocol lasts approximately 30 min) must be met. An important advantage of the Sports Athlete protocol compared to classic aerobic activities (e.g. running, cycling) is that this protocol has exercises for both lower and upper limbs, thus being a training protocol for the whole body, which can increase V˙E, HR, and V˙O2 values to exercise through a greater amount of muscle mass involved (Jensen-Urstad, Svedenhag & Sahlin, 1994).

When evaluating the protocol in phases, significant differences were found between average HR of the ‘warm-up’ and the ‘main phase.’ As desired, the intensity of the ‘warm-up’ period was lower than the intensity of the ‘main phase,’ which follows the recommendations for exercise prescription (Powers & Howley, 2017). This is a very important aspect, since the ‘warm-up’ may be related to the reduction of the probability of muscular injuries due to stretches or dislocations (Woods, Bishop & Jones, 2007) in addition to influencing subsequent physical (Fradkin, Zazryn & Smoliga, 2010) and muscular performance (Mascarin et al., 2015). On the other hand, a significant increase of the HR and V˙O2 occurred at the end of the ‘cool-down.’ This is contrary to the recommendations for exercise prescription (Pescatello et al., 2014; Powers & Howley, 2017), since the ‘cool-down’ should provide a gradual decrease of the HR (Powers & Howley, 2017). A possible explanation for the ‘cool-down’ intensity remaining high may be related to the Mountain Climbers exercise present in the ‘cool-down’ period, since the participants arrived very close to the ‘maximum effort’ (reported by participants), reflecting the amount of repetitions performed both in this exercise as in the others that constituted the ‘cool-down.’

Strengths and limitations of the study

To the best of our knowledge, this is the first study to evaluate the intensity of an exergame through the HR and V˙O2 values at VT1 and VT2 obtained from GXT. Most studies evaluated the intensity of exergames through percentages of the HRmax recommended by ACSM. The Sports Athlete protocol consists of exercises that people usually perform in their traditional routine of training (e.g. jog, bicycle crunches, push-ups, jump rope…), thus our results can be extrapolated to exercise facilities/gyms.

On the other hand, the participants in the present study performed all sessions coupled to the mouthpiece of the metabolic system. The discomfort generated by the salivation and fatigue of the chewing muscles (reported by the participants) may have interfered in the performance of the participants, although the performance of participants was better than suggested by the exergame. Thus, studies evaluating the influence of using the mouthpiece or mask over performance in exergames are needed, besides the accomplishment of the Sports Athlete protocol of the exergame Hollywood Workout in a non-laboratory context. Indeed, the study population included in this study were physically active. Therefore, future studies with sedentary individuals are needing to confirm our finding and understand the transferability of this findings to sedentary populations. Furthermore, the use of this protocol in the long term may be important to confirm the hypothesis that it would improve musculoskeletal and cardiorespiratory fitness, as well as possible changes in body composition as a function of energy expenditure. Blood lactate assessment could contribute to information on metabolic stress and anaerobic pathway contributions, since many participants reported high peripheral muscle fatigue. Nevertheless, we believe that these limitations do not prevent the conclusions of the study to be drawn.

Conclusions

From the present data, it appears that the exergames used in the present study can be classified as light to moderate physical exercise for the participants evaluated. Indeed, during the exergames trial, V˙O2 and HR remained predominantly below the VT1 level previously assessed during the GXT. Confirming this finding, the average V˙O2 and HR during the exergames were according to ACSM’s recommendations. Thus, exergames could be an interesting alternative to traditional forms of exercise as a tool for increasing physical fitness.

Supplemental Information

Supplemental Information 1 Raw data.

Click here for additional data file.

We would like to thank the participants for their effort and commitment to the research project.

Additional Information and Declarations

Competing Interests

Author Contributions

Human Ethics

Data Availability

The authors declare that they have no competing interests.

Ricardo B. Viana conceived and designed the experiments, performed the experiments, analysed the data, prepared figures and/or tables, authored or reviewed drafts of the paper, approved the final draft.

Rodrigo L. Vancini analysed the data, authored or reviewed drafts of the paper, approved the final draft.

Carlos A. Vieira analysed the data, authored or reviewed drafts of the paper, approved the final draft.

Paulo Gentil analysed the data, authored or reviewed drafts of the paper, approved the final draft.

Mário H. Campos analysed the data, authored or reviewed drafts of the paper, approved the final draft.

Marilia S. Andrade analysed the data, authored or reviewed drafts of the paper, approved the final draft.

Claudio Andre B. De Lira conceived and designed the experiments, performed the experiments, analyzed the data, contributed reagents/materials/analysis tools, authored or reviewed drafts of the paper, approved the final draft.

The following information was supplied relating to ethical approvals (i.e., approving body and any reference numbers):

The Federal University of Goiás granted Ethical approval to carry out the study within its facilities (Ethical Application Ref: 1.459.010).

The following information was supplied regarding data availability:

The raw measurements are provided in the Supplementary File.

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
