# Peer review of "Profiling exercise intensity during the exergame Hollywood Workout on XBOX 360 Kinect®"

_PeerJ, doi:10.7717/peerj.5574_

## Round 0.1 · original submission · Minor Revisions

Dear authors,

Your manuscript was evaluated by two expert reviewers. Please address reviewer´s concerns in detail.

Reviewer 1 ·

Basic reporting

This study aims to prove whether an exergame can be classified as a moderate form of exercise by running a series of experiment on 39 young male participants aged 25.9 +- 4.9.

This is the first attempt to evaluate the intensity of an exergame and whether it compares to traditional ways of exercising

The paper is well-written and well-structured. The style is also adequate. Most sections are clear and concise.

Very relevant work cited in the introduction and discussion sections. This suggests that the results obtained during the experimentation are in accordance with relevant studies done in the past.

I strongly believe this paper is a significant contribution, however, I have a few questions and concerns:

My main concern is that the chosen joystick-based game is not quite in alignment with the study design.

In your study, you exposed your participants to an exigent exercise in Phase 1 and a relatively demanding exergame in Phase 2. Then they play a fictional fighting game with a controller, not a sports game.

I think it would have made more sense to have them play a boxing exergame such as UFC Personal Trainer, or Kinect Sports Boxing, then have them played with the chosen fighting game.

The theme of a game can affect significantly your participants’ responses, feelings and attitudes towards the experiments. It important to understand that even traditional joystick-based games can also lead to increased physiological responses. Suspense intensive games such as the tactical shooter Swat 4 can easily increase the levels of adrenaline which directly affect heart rate, blood pressure, expansion of air passages of the lungs, the pupil in the eye, and even maximise blood glucose levels.

It would have made more sense to expose them to the same activity, but have them interact in different ways.

Also, I would like to know why all participants are males. Was there a criteria for excluding females from the study?

Aside from these, the paper is novel and the contribution is significant to the field of knowledge

Experimental design

N=39
Age 25.9 +- 4.9
Male only
Inclusion Criteria No to all items in PAR-Q
Participants performed exigent exercise
Then played an exergame (randomly allocated) (on a separate day)
Control played with a joystick (randomly allocated) (on a separate day)


The study design is very solid and in my opinion aligns well with the objectives of the study
It’s a bit unusual to use the term ‘control’ given that all participants went into intervention
Control refers to a group of participants that are also observed but are not part of the intervention.
I would probably suggest that this is removed.

Validity of the findings

Most results obtained during the experimentation are in accordance with relevant studies done in the past. This suggests that such findings are clinically meaningful.

Additional comments

See below some minor comments:

Abstract

Line 22-55 - I believe this line needs re-writing
What I get from this sentence is that you are comparing an exergame session with exergame session. Isn’t this the same?
Would not it make more sense to compare and exergame vs regular exercise vs a joystick-based game?

Introduction
Line 90 - Add Ref to GXT

Methods
Line 136 - Add photo of Participant performing the exercise

Line 159 - Add Screenshot of game and/or photo of Participant playing with the game

Reviewer 2 ·

Basic reporting

- The article has been written in English in a clear and precise manner. No major errors are observed.
- The introduction demonstrates sufficient background to understand the need to know the exercise intensity based on metabolic thresholds (first and second ventilatory thresholds (VT1 and VT2)) during the performance of exergames. There are only reports based on clinical and indirect cardiovascular variables. In addition, the introduction shows sufficient evidence in favor of the exergames and the approach to the research hypothesis.
- The manuscript is structured according to the standards of PeerJ, it has a well-organized abstract of 251 words. The manuscript includes all the results of the investigation.

Experimental design

- The manuscript complies with the Aims and Scope of PeerJ.
- The research question is well supported and defined: “It is important to note that none of these studies determined exercise intensity based on metabolic thresholds obtained in a continuous laboratory treadmill test, such as maximal graded exercise testing (GXT)”.
- The research complies with rigorous methodological and ethical criteria of the international research community. In general, it is a clear and precise method.

Validity of the findings

- Line 158: “This result corroborates those of Lanningham-Foster et al.(2006) and Mills et al.(2013) who reported HR and V ̇O2 values significantly lower in joystick rather than in exergame sessions. This can be explained by the lower recruitment of large muscle groups during practice of the joystick game, since the participants usually only move their hands and fingers”. This comparison (results of the present investigation with the reports of the years 2006 and 2013) and explanation can not be made, since the participants of the present research differ considerably in relation to age and type of physical activity / exergame. This paragraph must be deleted.
- Line 267 to 282: carry out a valid discussion and comparison between the results of the present investigation and the reports of previous years. Caution regarding the type of population, age, anthropometric characteristics and types of physical activity / exergame performed. The authors must make a meticulous and extrapolated comparison.
- The conclusion is related to the objective of the investigation.

Additional comments

- The research is novel and original since it tries to determine the intensity of exercise performed with exergames. This information is not present in the state of the art about exergames and partial immersion virtual reality.

---

## Round 0.2 · accepted · Accept

Congratulations on a well written manuscript!

Reviewer 1 ·

Basic reporting

I'd like to thank the authors of the manuscript for incorporating all the feedback received during the review stage and clarifying all the questions and concerns that were raised.

The latest version of the article reads very well and explains clearly what the study is about and its outcomes.

I was a bit critical of some claims and the study design, but after going through the rebuttal and revising the updated document, I strongly believe that this is a very significant contribution.

This work is very novel and, in my opinion, it meets the standards of this journal.
I would argue for this paper to be published.

Experimental design

NA

Validity of the findings

NA

Additional comments

I'd like to thank the authors of the manuscript for incorporating all the feedback received during the review stage and clarifying all the questions and concerns that were raised.

The latest version of the article reads very well and explains clearly what the study is about and its outcomes.

I was a bit critical of some claims and the study design, but after going through the rebuttal and revising the updated document, I strongly believe that this is a very significant contribution.

This work is very novel and, in my opinion, it meets the standards of this journal.
I would argue for this paper to be published.

Reviewer 2 ·

Basic reporting

no comment

Experimental design

no comment

Validity of the findings

no comment

Additional comments

The modifications made by the authors of the manuscript are related to the suggestions and comments made by me. I recommend to accept the manuscript.